# Plasma Levels of Pentraxin 3: A Potential Prognostic Biomarker in Urinary Bladder Cancer Patients

**DOI:** 10.3390/ijms25063473

**Published:** 2024-03-20

**Authors:** Anders Vikerfors, Sabina Davidsson, Jessica Carlsson, Tomas Jerlström

**Affiliations:** Department of Urology, Faculty of Medicine and Health, Örebro University, 703 62 Örebro, Sweden; sabina.davidsson@regionorebrolan.se (S.D.); jessica.carlsson@regionorebrolan.se (J.C.); tomas.jerlstrom@regionorebrolan.se (T.J.)

**Keywords:** urinary bladder cancer, plasma biomarker, diagnostic, prognostic, PTX3

## Abstract

Urinary bladder cancer (BC) represents a major health issue, and identifying novel biomarkers for early disease detection and outcome prediction is paramount. It has already been established that the immune system plays a role in tumour initiation and progression in which the inflammatory marker pentraxin 3 (PTX3) might be involved, presenting a variety of functions in different cancers. The aim of this study was to investigate whether plasma levels of PTX3 could be used as a biomarker for patients with BC. Plasma levels of PTX3 were determined in 118 BC patients and 50 controls by ELISA. Patients with BC had significantly higher PTX3 levels compared to controls. The value as a diagnostic biomarker is probably limited, however, since no significant difference in PTX3 levels was seen between patients with non-muscle-invasive BC and controls; they were seen only between patients with muscle-invasive disease and controls. However, the potential value of PTX3 as a prognostic biomarker was indicated by significantly higher PTX3 levels in patients who developed metastatic disease during follow-up compared to patients who did not develop metastatic disease. The conclusions from this study are that plasma levels of PTX3 have limited value as a diagnostic biomarker, although they have potential as a prognostic biomarker for patients with BC.

## 1. Introduction

Urinary bladder cancer (BC) is the ninth most common cancer worldwide and the thirteenth deadliest [1]. The majority of cases are low-grade and non-muscle-invasive bladder cancers (NMIBCs) at presentation [2] with a relatively good prognosis. However, approximately 15% of NMIBCs progress to a muscle-invasive bladder cancer (MIBC), which is more aggressive and associated with worse prognosis. If untreated, the 5-year survival rate for these patients is low (20.9%), and it is even lower with the presence of visceral metastases (6.8%) [3]. The most common type of BC in developed countries is urothelial carcinomas, although some rare histological subtypes exist, including squamous, glandular, or sarcomatoid cell features. The high incidence and recurrence rate of BC, demanding long-term patient follow-up, makes BC one of the most expensive malignancies to manage from a health-economic perspective [4]. In Sweden, BC survival rates have improved in the past few decades; however, incidence has increased and mortality remains relatively unchanged [5]. Considering these aspects, BC represents a major public health issue, and identifying novel ways to detect and predict disease progression in NMIBC is paramount. Thus, there is an urgent need for improved biomarkers to help identify the disease earlier, risk-stratify patients, improve prognostication, and predict patient outcomes.

It is well established that the body’s immune defence plays an important role in the initiation and progression of cancer. The interest in studying inflammatory-associated biomarkers in BC has increased over the last decade. Studies investigating, for example, the systemic immune-inflammation index (SII), neutrophil-to-lymphocyte ratio (NLR), and the urinary microbiome, have shown promising results as biomarkers for BC [6,7,8,9]. However, further research identifying other candidate biomarkers is still needed. Pentraxins are a superfamily of soluble proteins that are essential components of the humoral innate immune response. This family of proteins can be divided into long and short pentraxins, where C-reactive protein (CRP) and serum amyloid P component protein (SAP) are considered to be prototypes for the short pentraxins [10]. The prototype for long pentraxins, Pentraxin 3 (PTX3), is a multifunctional protein that plays an essential role in inflammation, complement activation, and vascular remodelling [11,12]. Unlike CRP, which is produced by hepatocytes, PTX3 is produced in a wide range of cells at the site of inflammation, including endothelial and epithelial cells, fibroblasts, and monocytes [12].

Even though there is a well-established link between inflammation and PTX3, the role of PTX3 in carcinogenesis is not yet fully understood [13]. In BC, patients with NMBIC have been found to have higher PTX3 expression in tumour tissues compared to patients with MBIC, suggesting that the overexpression of PTX3 could be associated with a better prognosis in BC patients [13]. Circulating levels of PTX3 have also been investigated as a biomarker previously. Increased PTX3 serum levels have been found in patients with various types of cancer compared to healthy controls [14], and high serum levels of PTX3 have also been associated with a worse prognosis [14,15,16]. The potential of using circulating PTX3 as a biomarker for BC has been investigated previously by Goodison et al. In their study, no difference was found when comparing urinary levels of PTX3 in patients with newly diagnosed BC and BC-free patients [17]. However, to the best of our knowledge, no studies have been performed measuring circulating PTX3 in blood samples from BC patients. The aim of this study was thus to investigate whether plasma levels of PTX3 could be used as a potential biomarker to identify the disease earlier and predict outcomes for patients with BC.

## 2. Results

A total of 118 BC patients and 50 participants without BC (controls) were recruited between October 2016 and March 2019. The median follow-up time was seven months (range 0–30 months). Patients diagnosed with BC were significantly older, and more often men, compared to the controls (*p* < 0.05). Furthermore, BC patients were more frequently smokers or previous smokers compared to the controls. No difference in body mass index (BMI) was evident between the two groups. Patient characteristics are described in Table 1.

### PTX3 and Clinical Variables

Pentraxin 3 was detectable in 167 of 168 (99.4%) of the available plasma samples, and the median concentration of PTX3 was 0.73 ng/mL (InterQuartile Range (IQR): 0.86). No significant differences in PTX3 levels were seen between genders (*p* = 0.893) or between smoking status categories (never, former, current smoker, *p* = 0.529). Patients with BC had significantly higher PTX3 levels compared to controls (*p* = 0.021); the median levels of PTX3 were 0.83 ng/mL and 0.60 ng/mL, respectively (Figure 1). Logistic regression was performed in order to further investigate the role of PTX3 as a diagnostic biomarker for BC. A one-unit increase in PTX3 levels was associated with 1.53 times higher odds of BC; however, this association was not statistically significant (OR: 1.53, 95% CI: 0.96–2.44). Adjusting the model for gender, smoking, age, and BMI did not change this result (OR: 1.51; 95% CI: 0.89–2.54).

PTX3 levels did not differ between G1-G3 grades of the tumour, or when comparing low-grade tumours to high-grade tumours (*p* = 0.123 and *p* = 0.071, respectively, Figure 1 and Table 2). Patients with MIBC (n = 68) had significantly higher levels of PTX3 in plasma compared to NMIBC patients (n = 50) (*p* < 0.001). A one-unit increase in PTX3 levels was associated with 4.24 times higher odds of MIBC compared to NMIBC (95% CI: 1.84–9.77). This association remained statistically significant when adjusting the model for gender, smoking, age, and BMI (OR: 3.72, 95% CI: 1.54–8.99). However, no significant difference in PXT3 levels was found between patients with NMIBC and controls (*p* > 0.05). A portion of the patients (48.5%) were included in the study after referral from other regions for consideration for cystectomy surgery. These patients had undergone a transurethral resection of a bladder tumour (TURBT) before inclusion, which could have had an effect on plasma levels of PTX3. Hence, a subgroup analysis was performed to compare PTX3 levels among MIBC cases that were included before the first TURBT (n = 64) and MIBC cases included after a TURBT (before a cystectomy, n = 54). We found no difference in PTX3 levels between the groups (*p* = 0.558).

Patients who developed metastatic disease during follow-up (n = 16) had significantly higher PTX3 plasma levels at baseline compared to patients without metastatic progression during follow-up (*p* = 0.009, Table 2). A one-unit increase in PTX3 levels was associated with 1.27 times higher odds of developing metastatic disease; however, this was not statistically significant (OR: 1.27, 95% CI: 0.97–1.65). Adjusting the model for pT stage and histological subtype did not change this association (OR: 1.27, 95% CI: 0.93–1.72). No significant difference in PTX3 levels was seen between patients with primary metastatic disease and patients who developed metastatic disease during follow-up (*p* = 0.538).

The majority of tumours were classified as urothelial tumours; however, twelve patients had tumours with mixed histological features of squamous epithelial differentiation, and an additional three patients had tumours with sarcomatoid features. When comparing PTX3 levels between tumours of different histological subtypes, there was a trend towards higher PTX3 levels among patients with tumours with elements of squamous epithelial histology and tumours with sarcomatoid features compared to patients with tumours of pure urothelial histology (*p* = 0.053; Figure 1 and Table 2).

## 3. Discussion

Cancer in the urinary bladder is a disease with diverse clinical outcomes, depending on several factors, including tumour stage, grade, and overall patient health status. Today, the initial examination of suspected BC and subsequent disease monitoring involve cystoscopy, imaging, and histopathological evaluation. Frequent cystoscopy examinations and repeated imaging make BC one of the most resource-consuming malignancies worldwide. In order to reduce adverse effects of repeated cystoscopies (e.g., infections) and side effects of surgery, and to lower health care costs for this group of patients, novel non-invasive modalities to detect, predict, and monitor BC are needed.

Elevated levels of circulating PTX3 have been observed in several malignancies, including colorectal, lung, and hepatocellular cancers [14,15,16]. The results from the present study show that patients with BC also have significantly higher PTX3 levels compared to patients without the disease (controls). However, the value of PTX3 as a diagnostic biomarker for BC may be questionable since significant differences were observed between controls and patients with MIBC, and between patients with NMIBC and MIBC, although not between controls and patients with NMIBC. A possible reason for the lack of difference in PTX3 levels between controls and patients with NMIBC may be the choice of controls. Patients in the control group were included after referral to the urology clinic due to gross haematuria, where they underwent urography and cystoscopy, clearing patients from BC. Often, benign findings such as haemorrhagic cystitis, stones in the urinary tract, or indwelling catheters cause gross haematuria. These benign findings could theoretically contribute to an activation of the immune system, leading to increased PTX3 levels in the control group as well. The clinical value of a diagnostic biomarker for BC would be of most use in a patient group with risk factors and alarm symptoms such as gross haematuria, hence the choice of controls in the present study. Another possibility for the lack of difference in PTX3 levels between controls and patients with NMBIC could be due to the absence of tumour invasion into the bladder walls. Since the tumour did not invade the bladder muscle, there is a possibility that the tumour was not large enough to have an effect on systemic levels of PTX3. Based on the lack of a significant difference in PTX3 levels between controls and patients with NMIBC within this study, PTX3 levels do not seem to have potential as a diagnostic biomarker for BC. This is in line with previous studies, where PTX3 levels were measured in urine from BC patients and disease-free controls and no significant differences between the two groups were found [17,18]. Unfortunately, no validated ELISAs for measuring PTX3 in urine samples were available when this study was performed, thus we did not attempt to measure PTX3 in urine in the present study.

In the present study, we found that patients with MIBC had significantly higher PTX3 plasma levels compared to patients with NMIBC. However, opposing results concerning the function of PTX3 in BC have been reported. Matarazzo et al. recently demonstrated lower PTX3 expression in MIBC tissue compared to NIMBC tissue, thus portraying PTX3 as an oncosuppressor associated with less advanced disease in BC [19]. One important distinction between these two studies is that Matarazzo et al. investigated the expression of PTX3 in tumour tissue from the urinary bladder while, in the present study, we investigated systemic levels of circulating PTX3 in plasma. It should be highlighted that circulating PTX3 levels do not exclusively reflect PTX3 expression within the urinary bladder, thus comparing results in the present study to results presented by Matarazzo et al. is difficult.

The results in this study showed that patients who developed metastatic disease during follow-up had significantly elevated PXT3 levels at time of inclusion compared to patients without disease progression. These results are in line with previous findings demonstrating an association between increased serum PTX3 levels and metastatic disease in patients with renal cell carcinomas [20] and in patients with colorectal cancer [21]. Previous studies have also demonstrated differential levels of circulating PTX3 between histological subtypes of gliomas and lung cancer [22,23]. In the present study, no significant differences in PTX3 levels between histological subtypes could be found, which could be due to the low number of cases with each histological subtype. However, BC patients with histological elements of sarcomatoid or squamosal cell differentiation had higher median levels of PTX3 (*p* = 0.053) compared to patients diagnosed with pure urothelial carcinoma tumours. Both the sarcomatoid and squamous differentiations are aggressive subtypes of BC, which are often muscle invasive upon diagnosis and with a high propensity to metastasize [24,25]. This is also the case in the present study, where 14 out of 15 patients diagnosed with sarcomatoid or squamous cell differentiation had MIBC upon diagnosis. The results from the present study indicate that elevated plasma levels of PTX3 are associated with a more advanced Tumor, Node, Metastasis (TNM) stage in BC patients, and thus have potential as a biomarker for the prognostication of the disease. In a clinical setting, high plasma levels of PTX3 could potentially serve as a valuable biomarker for stratifying BC patients based on their disease severity and prognosis. Patients with elevated PTX3 levels may represent a subgroup requiring early, more aggressive treatment strategies or additional diagnostic imaging to detect any potential tumour progression at an earlier stage. By identifying these high-risk individuals, clinicians can intervene promptly, potentially preventing disease advancement and improving overall patient outcomes. Therefore, integrating PTX3 measurement into routine clinical assessments could potentially enhance the management and personalized care of BC patients. However, further studies are needed to evaluate this potential clinical use of PTX3.

The present study has some strengths and limitations. One of the strengths is that demographic characteristics from the study participants correlate well with previous international and national epidemiological BC data [5,26,27]. Another strength of this study is the choice of controls, which reflects a clinical setting where the major aim is to distinguish between patients presenting with gross haematuria due to BC and those presenting the same condition due to other causes. A limitation of the study is mainly the small cohort size, with small numbers of patients in each category of risk group, histological subtypes, and metastatic disease. The small cohort size also limits the possibility of generalizing the results to a broader population. Another limitation is the limited follow-up time (median of 7 months), which limits statistics, including follow-up. Longer follow-up time is necessary in order to fully elucidate the prognostic value of PTX3. Even though the study was underpowered for some of the analyses, the results suggest a potential value of circulating PTX3 levels as a prognostic biomarker for BC. In this study we did not investigate any potential biological mechanisms underlying the association between PTX3 levels and BC progression. Furthermore, the source of circulating PTX3 is unknown, which could be considered a limitation.

## 4. Materials and Methods

### 4.1. Study Subjects

In the present study, patients were recruited from the BLAdder cancer Blood and Urine Study (BLABUS). BLABUS is a prospective cohort study where patients with suspected BC, often due to macroscopic haematuria, are referred to the Department of Urology, University Hospital of Örebro, Sweden. The vast majority of the patients underwent computed tomography urography and an outpatient cystoscopy examination. Patients with a visual sign of a bladder tumour at cystoscopy or urography thereafter underwent a transurethral resection of the bladder tumour (TURBT). The diagnosis of BC was confirmed by histopathological examination of the resected tissue samples. Tumour stage was defined according to the 2009 TNM classification of BC, and tumour grade was designated according to the WHO 2004 and WHO 1999 grading schemes. Patients without a history of BC and with a normal cystoscopy/urography examination were included as controls. A portion of the patients (45.8%) was included after referral from other regions for consideration of cystectomy surgery. Clinical data from the patients were extracted from medical records until 1 May 2019. Cause of death was defined as cancer-specific if the patient had metastatic BC at time of death, and unrelated to BC if no sign of metastatic BC was detected at time of death.

The study was approved by the ethical committee of the Uppsala/Örebro region (approval number: 2016/088). All patients signed a written consent to participate in the study.

### 4.2. PTX3 Measurement

At inclusion, blood samples were obtained from all patients. Blood samples were collected in EDTA tubes, and plasma was obtained by centrifugation at 3000× *g* for 10 min at 4 °C. Plasma samples were aliquoted and stored at −80 °C until analysed. All samples were obtained prospectively, and samples from cases and controls were processed simultaneously.

Plasma levels of PTX3 were measured with a commercially available ELISA kit, according to the manufacturer’s instructions (R&D Systems Inc., Minneapolis, MN, USA; Catalogue no. DPTX30). Control samples with known concentrations of PTX3 (low, medium, and high) were included in each ELISA (R&D Systems Inc., Minneapolis, MN, USA; Catalogue no. QC151). The minimum detectable level of PTX3 in the ELISA was 0.116 ng/mL, and the detection range was 0.3–20 ng/mL. The optical density was measured using a Multiskan Ascent plate reader (Thermo Fisher Scientific, Waltham, MA, USA) at 450 and 540 nm.

All samples were measured in duplicate and, for each sample, the coefficient of variation (CV, %) was calculated. The acceptable range of CV was 0–15%; samples with higher CV values were re-analysed. Blanks and standards were assayed according to the manufacturer’s instructions. The mean values of absorbance vs. concentrations were plotted, and a log–log curve fit was applied. R^2^ values above 0.9 were considered acceptable.

### 4.3. Statistical Analyses

A Shapiro–Wilk’s test was used to test data for normality. Mann–Whitney U-tests or Kruskal–Wallis tests were subsequently performed to investigate PTX3 levels between groups of patients. Logistic regression models were used to estimate odds ratios (ORs) and 95% confidence intervals (95% CIs) of the association between PTX3 levels and metastatic disease, adjusting for pT stage and histological subtype. All analyses were performed using SPSS Statistics version 25 (IBM SPSS, New York, NY, USA). Two-tailed *p*-values < 0.05 were considered statistically significant.

## 5. Conclusions

This is the first study to investigate PTX3 plasma levels in patients with BC and control participants with gross haematuria. The results indicate that circulating PTX3 levels hold little potential as a diagnostic biomarker for BC. However, high PTX3 levels were associated with a more aggressive disease, as indicated by tumour stage and the development of metastatic disease, and thus circulating PTX3 levels could be a potential biomarker for prognostication of BC. By identifying patients with high-risk BC, it could be possible to prevent tumour progression by applying an aggressive therapeutic approach early on and thus obtain a more favourable clinical outcome for these patients. However, further prospective studies are needed to assess the use of plasma levels of PTX3 as a biomarker for monitoring of BC patients.

## Figures and Tables

**Figure 1 ijms-25-03473-f001:**
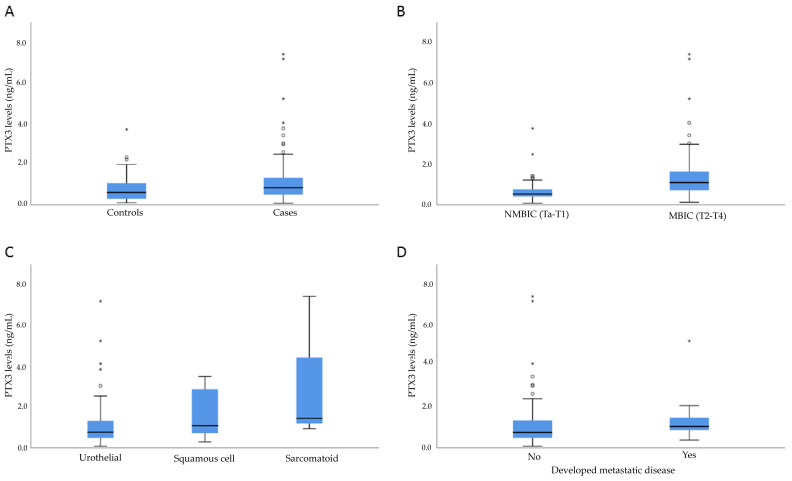
Pentraxin 3 (PTX3) levels in plasma samples. (**A**) Across cases and controls. (**B**) Patients with non-muscle-invasive bladder cancer (NMIBC) versus patients with muscle-invasive bladder cancer (MIBC). (**C**) Across three histological subtypes of urinary bladder cancer. (**D**) Patients who developed metastatic disease during follow-up versus patients without development of metastatic disease. Circles = outliers and stars = extreme outliers.

**Table 1 ijms-25-03473-t001:** Patient demographics of 118 patients with bladder cancer (BC) and 50 participants without BC (controls).

	Controls (n = 50)	BC (n = 118)	*p*
Age, median (IQR)	68.5 (18.1)	74.6 (8.3)	0.001
Gender, n (%)			0.007
Men	27 (54)	89 (75.4)	
Women	23 (46)	29 (24.6)	
BMI, median (IQR)	25.1 (6.7)	25.9 (4.7)	0.852
Smoking, n (%)			0.006
Yes	5 (10.0)	21 (17.8)	
No	35 (70.0)	51 (43.2)	
Former	10 (20.0)	46 (39.0)	

IQR = InterQuartile Range. BMI = Body Mass Index. Minimum and maximum values of PTX3 in each category is found in Appendix A.

**Table 2 ijms-25-03473-t002:** Pentraxin 3 (PTX3) levels in plasma and tumour characteristics.

	BC (n = 118)	PTX3 (ng/mL) Median (IQR)	*p*
Invasiveness, n (%)			<0.001
NMIBC	50 (42.4)	0.52 (0.39)	
MIBC	68 (57.6)	1.08 (0.95)	
Metastatic disease, n (%)			0.009
No	90 (81.8)	0.73 (0.84)	
Developed metastatic disease *	16 (14.5)	1.02 (0.72)	
Primary metastatic disease	4 (3.6)	1.35 (5.13)	
Missing	8		
Tumour grade, n (%)			0.123
Grade 1	13 (12.0)	0.47 (0.38)	
Grade 2	25 (23.1)	0.81 (0.91)	
Grade 3	70 (64.8)	0.87 (0.81)	
Missing (CIS)	10		
Tumor grade, n (%)			0.071
Low grade	22 (20.4)	0.51 (0.87)	
High grade	86 (79.6)	0.87 (0.83)	
Missing (CIS)	10		
Histological subtype, n (%)			0.053
Urothelial carcinoma	103 (87.3)	0.74 (0.81)	
UC with squamous cell features	12 (10.2)	1.05 (2.28)	
UC with sarcomatoid features	3 (2.5)	1.4 (-)	

NMIBC—non-muscle-invasive bladder cancer. MIBC—muscle-invasive bladder cancer. CIS—carcinoma in situ. UC—urothelial carcinoma. * Developed metastatic disease during follow-up. Primary metastatic disease—lymph node metastases found during cystectomy.

## Data Availability

The data presented in this study are available on request from the corresponding author.

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
