# Peer review of "Plasma Levels of Pentraxin 3: A Potential Prognostic Biomarker in Urinary Bladder Cancer Patients"

_ijms, 2024, doi:10.3390/ijms25063473_

Round 1

Reviewer 1 Report

Comments and Suggestions for Authors

The article titled "Plasma levels of Pentraxin 3: A potential prognostic biomarker in urinary bladder cancer patients", investigates the potential of Pentraxin 3 (PTX3) plasma levels as a biomarker for urinary bladder cancer (BC). The study is significant as it addresses a major health issue, aiming to find novel biomarkers for early disease detection and outcome prediction. It highlights the role of the immune system, specifically PTX3, in tumor progression and its potential as a prognostic biomarker in BC. The research demonstrates that while PTX3 has limited value as a diagnostic biomarker, it shows promise for prognostication, especially in identifying patients at higher risk of developing metastatic disease.

Interest and Novelty:

  • The study's novelty lies in its focus on PTX3 plasma levels in BC patients, an area not extensively explored before. It contributes to the existing literature by providing new insights into the potential utility of PTX3 as a prognostic biomarker.

Areas for Improvement:

  • Sample Size and Diversity: The cohort size could be expanded to increase the study's power and include a more diverse patient population to generalize the findings. if this is not probìssibile this should be mìbetter liste din the study limitation section. 
  • Long-term Follow-up: Extending the follow-up period would provide more comprehensive data on PTX3's prognostic value over time. if this is not probìssibile this should be mìbetter liste din the study limitation section. In addition, try to suggest further research that could elucidate the biological mechanisms underlying the association between PTX3 levels and BC progression.
  • Comparative Analysis & Discussion: Comparing PTX3 with other biomarkers could help position its utility in the context of existing diagnostic and prognostic tools. To further underscore the potential of identifying novel biomarkers for BC, recent research exploring the urobiome has revealed specific bacterial signatures associated with BC patients, offering new avenues for non-invasive diagnostics and prognostics (PMID: 38298766). This study's findings on the abundance of Porphyromonas somerae in BC patients, particularly males over 50, align with our investigation into PTX3 levels, suggesting a multifaceted approach to BC biomarker discovery could enhance patient management and outcome prediction. In addition to our findings on PTX3, emerging evidence on systemic immune-inflammatory markers, such as the systemic immune-inflammation index (SII), underscores their prognostic significance in bladder cancer. A recent study demonstrated that higher preoperative SII values were associated with adverse oncological outcomes in patients undergoing radical cystectomy (PMID: 38138166). This further highlights the pivotal role of the immune system in bladder cancer progression and the potential of integrating various immune-related biomarkers for enhanced patient stratification and management.
  • Clinical Utility: Future studies should aim to translate these findings into practical applications, determining how PTX3 measurement can be integrated into clinical practice to improve patient outcomes.

In conclusion, the article presents a promising avenue for BC prognostication but calls for further studies to validate PTX3's utility and explore its integration into clinical workflows.

Author Response

We thank the reviewer for valuable comments and have attached a letter of response with the changes made. 

Reviewer 2 Report

Comments and Suggestions for Authors

I would congratulate with the authors for their work and for addressing an important topic. However, some points warrant mention:

1.        An English form revision of both the Abstract and the main text is required.

2.        In the background of the need for biomarkers for the early detection of BC, I suggest to mention in the “Introduction” section the role of previously identified markers such as in PMID: 38138166.

3.        In the “Introduction” section, lines 61-72 report redundant information. I suggest to revise the form and better reporting information.

4.        In both the “Results” section and the “Table 1”, results are not reported as median (IQR). Please check and correct.

5.        In the “Discussion” section, lines 146-148, this statement is not correct after seeing the results. If PTX3 levels differ “only” between patients with MIBC and NMIBC, this marker could never replace the role of cystoscopies in the follow-up of all BC.

6.        In the “Discussion” section, lines 149-171, I do not really agree with these paragraphs. The authors question the rise of PTX3 in benign conditions with haematuria, but I think that the real motivation behind the absence of differences in blood PTX3 levels between NMIBC and controls is to research the absence of the invasion of bladder walls. Indeed, levels are high in patients with MIBC. I suggest to the authors to explore this possibility, or to compare the reported data with the related literature (which is missing in these sentences).

7.        I suggest to run further statistical analysis (i.e., regression models) to assess the role of PTX3 in BC.

8.        In the “Discussion” section, lines 196-198, since the authors reported the PTX3 levels do not differ among tumour grades, this statement is not correct.

Comments on the Quality of English Language

Extensive English forma revision required.

Author Response

We thank the reviewer for the suggested changes and have attached a response letter where we have adressed these changes. 

Round 2

Reviewer 1 Report

Comments and Suggestions for Authors

the authors provided a comprehensive revised version. I consider it worth of publication.

Reviewer 2 Report

Comments and Suggestions for Authors

Thanks for providing such a comprehensive review of the mnuscript.